# Pollution and Risk Assessment of Polycyclic Aromatic Hydrocarbons in Urban Rivers in a Northeastern Chinese City: Implications for Continuous Rainfall Events

**Guangyi Mu [1], Dejun Bian [1],\*, Min Zou [2],\*, Xuege Wang [1] and Fangfang Chen [3]**

[1] Jilin Provincial Key Laboratory of Municipal Wastewater Treatment, Changchun Institute of Technology, Changchun 130012, China

[2] Key Laboratory of Vegetation Ecology, Ministry of Education, Institute of Grassland Science, Northeast Normal University, Changchun 130024, China

[3] Key Laboratory of Virtual Geographic Environment of Education Ministry, Nanjing Normal University, Nanjing 210046, China

\* Correspondence: ccgcxybiandj@163.com (D.B.); zoum658@nenu.edu.cn (M.Z.)

**Abstract:** Most studies have concentrated on the distribution of polycyclic aromatic hydrocarbons (PAHs) in air, water, and soil; however, little is known about their behavior during urban wet deposition. During frequent urban rainstorms, surface runoff carries large amounts of organic pollutants into water, which has a significant impact on the quality of the water environment. "Poseidon" made landfall in Jilin Province on 27 August, 3 September, and 7 September 2020, respectively, causing some impact on the surface water. Gas chromatography–mass spectrometry (GC–MS) was used to measure the concentrations of 16 major PAHs in stormwater and water samples from the Yitong River. The PAH concentrations in rainwater runoff gradually decreased with increasing rainfall duration. The PAHs in the Yitong River water samples were mainly composed of two to three rings, with total concentrations ranging 279.19–756.37 ng/L. Ratio characterization and principal component analysis of the sources of PAHs in the water samples revealed that some water bodies had also been contaminated by oil spills in addition to combustion emissions from coal and motor vehicle oil. Health and environmental risks were assessed for PAHs in water bodies from the Yitong River, and health risk assessment showed that exposure to PAHs in the water is not a health risk for human beings; however, the risk for children is higher than that for adults and requires attention. Although the environmental risk is moderate, the risk from Benzo(a)anthracene(BaA) alone is high and requires ongoing attention. In terms of the sustainability of drinking water sources, consideration should be given to taking necessary regulatory and protective measures to prevent further contamination. These results serve as a guide for managing PAHs in urban water bodies and managing water pollution.

**Keywords:** polycyclic aromatic hydrocarbons; typhoons; Yitong River; source analysis; boulder risk evaluation

## 1. Introduction

Polycyclic aromatic hydrocarbons (PAHs) are common organic pollutants, consisting of fused benzene rings [1,2]. Human activities, including emissions from inadequate fuel combustion, oil spills, and traffic emissions, have increased the sources of PAHs in Earth subsystems [3,4]. PAHs are hydrophobic and have a high octanol–water partition coefficient, increasing with the number of benzene rings; the more lipid soluble they are, the less water soluble they are, and the longer they are present in the environment [5]. Considering their hazardous effects on the environment and human health, the United States Environmental Protection Agency (USEPA) classified 16 PAHs in a priority list [6]. From a public health and environmental protection perspective, PAH pollution in aquatic

environments is of great concern because of its high toxicity to aquatic organisms and potential carcinogenicity upon consumption [7]. The sources, occurrence, transport, and fate of PAHs in the aquatic environment have been extensively studied [8–10].

Aquatic contamination has become a global issue in recent decades as a result of the industrial and agricultural development of organic compounds, such as PAHs [11]. China has achieved rapid economic and agricultural intensification, and is becoming increasingly aware of these environmental issues, especially in response to water quality degradation. Seven PAHs among the sixty-eight priority water pollutants have been proposed in China since 1990. In recent years, numerous studies have been carried out on the concentration levels and potential sources of PAHs in the Yangtze [12], Yellow [13–15], Songhua [16,17], and Pearl [18] rivers of China. While the proportion of water quality sections (categorized as I to III) nationwide has increased by 1.5%, the ecological and environmental quality has also significantly improved (China Environmental Statistics Bulletin, 2021). However, the water quality of rivers entering the sea is still slightly polluted [19], indicating that there are still significant environmental issues that need to be addressed. The 2021 China Environment Statistics Bulletin shows that although the country's ecological environment has improved significantly, rapid economic development and high energy consumption, along with the extensive human activities associated with it, such as industrial and agricultural activities and urbanization, have resulted in an ever-increasing discharge of pollutants into rivers. It can be noted that more previous studies paid more attention to the large rivers or lakes [20–24]. In fact, the most industrialized and urbanized cities have been and still are draining to the rivers through normal flow or run-off transport episodes. As a result, urban rivers suffer from an abundance of contaminants from anthropogenic activities which are chemically stable and difficult to decompose, such as PAHs.

Urban rivers constitute a significant portion of water and are vital to human consumption and the development of cities. With the accelerated urbanization process, rivers are receiving increasing terrestrial loadings with a significant increase in sewage discharge and non-point source pollution. In riverine waters, PAHs can enter through several different pathways, such as airborne particles, wastewater discharge, surface runoff, soil runoff, and oil leaks. PAHs accumulated in forest–grassland soils are easily transferred to rivers via surface runoff and groundwater. Although the number of impervious layers increased with the expansion of urban areas, PAHs were still enriched in impervious layers [25]. Notably, atmospheric wet/dry deposition and air–water exchange are important pathways for the transport of PAHs [26,27]. Furthermore, low to moderate disturbances caused by rainfall can lead to the release of PAHs due to resuspension [28]. Typhoons, which are storms over tropical oceans, bring rainfall. In certain episodic events such as typhoons and hurricanes, such storms are composed of unusually strong moist and hot oceanic air masses, and where typhoons pass through torrential rainfall cascades, which can reach hundreds of millimeters at a time, sometimes up to 1000 mm or more, the results are highly disruptive. The floods that accompany them can carry land-based materials from rivers into the sea, so typhoons can alter biogeochemical processes and affect biological productivity. For example, Typhoon Kalmadge, in 2008, during which southwestern Taiwan was greatly affected, provided an opportunity to explore this process, with millions of tons of suspended matter being discharged from the Gaoping River every day, and, for some days after the typhoon, suspended particulate matter continued to be discharged at a greater rate. Researchers explored the distribution and fate of PAHs in the Gaoping River [29–31], and found a dearth of the potential effects of PAHs after typhoons, particularly in urban rivers.

Changchun is located in Jilin Province in northeastern China, a cold, semi-arid climate that marks the formation of tropical cyclones as difficult. Three super typhoons in 2020, Typhoon 8, Typhoon 9, and Typhoon 10, entered the northeast via the Korean peninsula and remained relatively stable during this period due to the meridional circulation of the substratum and the northeastern cold vortex. All three typhoons were guided northwards along a similar path as the longitude line, which was "straight". In the case of Typhoon Bavet, for example, as it made its way northwards, dry air was constantly involved in its

circulation, adversely affecting its strength development, as is common with northward-moving typhoons. As the latitude of the typhoon rises, the thermal conditions on the sea surface deteriorate and the dry and cold air from the north gradually strengthens, weakening the typhoon's intensity, a feature also seen with Poseidon and Methac. However, the impact of the wind and rain had another variation. Typhoon Metsak entered the high latitudes and combined with cold air to become a temperate cyclone, with a wider impact and stronger rainfall and local transient winds. In Jilin, Heilongjiang, northeastern Inner Mongolia, and some parts of Liaoning, gales of 7 to 8 were widespread, with 9 to 11 in some areas; 49 national observation stations in Jilin and Heilongjiang had daily rainfall exceeding the historical extreme in September. Precipitation in the northeast was more than one time higher than normal for the same period, and more than four times higher in southern Heilongjiang and central and western Jilin. Typhoon "Poseidon" entered China's Jilin and Heilongjiang provinces on 8 September, with a high degree of overlapping rain and wind impacts with the previous period. From historical data, the average annual impact of typhoons in the northeast of China is about 1.2, but the "triple whammy" in just half a month is the first time since records began. The three typhoons landed in Changchun on 27 August, 3 September, and 7 September, causing a cumulative average precipitation of 237.9 mm, more than four times the annual normal, while the average daily rainfall in Jilin Province in 2019 was only 29.6 mm. The strong winds, intense rainfall, and long duration resulted in a high degree of overlap in hydrological and biogeochemical processes in urban rivers.

Regarding the fate and ecological risk of PAHs in urban rivers before and after typhoon transit, we collected the surfaced water samples from urban rivers of Changchun city. The specific aims of this study are (1) to analyze the 16 priority PAH levels' distribution in urban river; (2) to analyze the correlations between PAHs and water quality parameters; (3) to identify the sources of PAHs in rivers; (4) to assess ecological and human health risks of PAHs, considering toxic equivalents (TEQs). With a view to filling an important knowledge gap concerning the effects of extreme rainfall events on PAHs in urban rivers, we hope to provide some scientific references for the prevention and controlling of organic pollutants in the studied area and other similar urban waters.

## 2. Materials and Methods

### 2.1. Location of Study

Changchun (43°05′–45°15′ N, 124°18′–127°05′ E), located in the Songliao Plain, is the natural geographical center of the northeast China (Figure 1). Changchun City has a resident population of 9,087,200 and an area of 24,592 km². The study city is located in the continental monsoon zone, with an average annual temperature of 5.3 °C and a precipitation of 583.5 mm (precipitation occurs between June and September). It is also characterized by a dry and windy spring from April to May, with an average wind speed of 3.9 m/s and an extreme maximum wind speed of 30 m/s. The Yitong River is the only river that flows through the province hinterland; it is 342.5 km long, with a 23 km portion across the city. Changchun continuously releases domestic and industrial wastewater into the Yitong river. In earlier investigations [32,33], the water quality in the Yitong River was poor, and some of its tributaries have deteriorated into sewage ditches. Despite the significant industrialization of this city, the urban region is environmentally vulnerable and heavily polluted with PAHs.

### 2.2. Instruments and Reagents

Gas chromatography–mass spectrometry (GC–MS, TSQ8000Evo), solid-phase extraction (SPE) system (Cleanert S C18-SPE, Agela Technologies Inc., Tianjin, China), K-D concentrator, nitrogen blowing apparatus (N-EVAP-24), and analytical balance with 0.01 g resolution were used. Reagents and materials—that is, dichloromethane, hexane, methanol, acetone, isopropanol, 1:1 ($v/v$) methanol/aqueous solution, n-hexane, Sodium hyposulfide, anhydrous sodium sulfate, potassium dichromate-concentrated sulfuric acid wash, Milli-Q

water, 500 mg/3 mL C18 (6 mL; Agela Technologies Inc., Torrance, CA 90501, USA), Whatman glass fiber membrane (0.7 μm), nitrogen (N$_2$), and helium (He) (purity > 99.99%)—were used in this study.

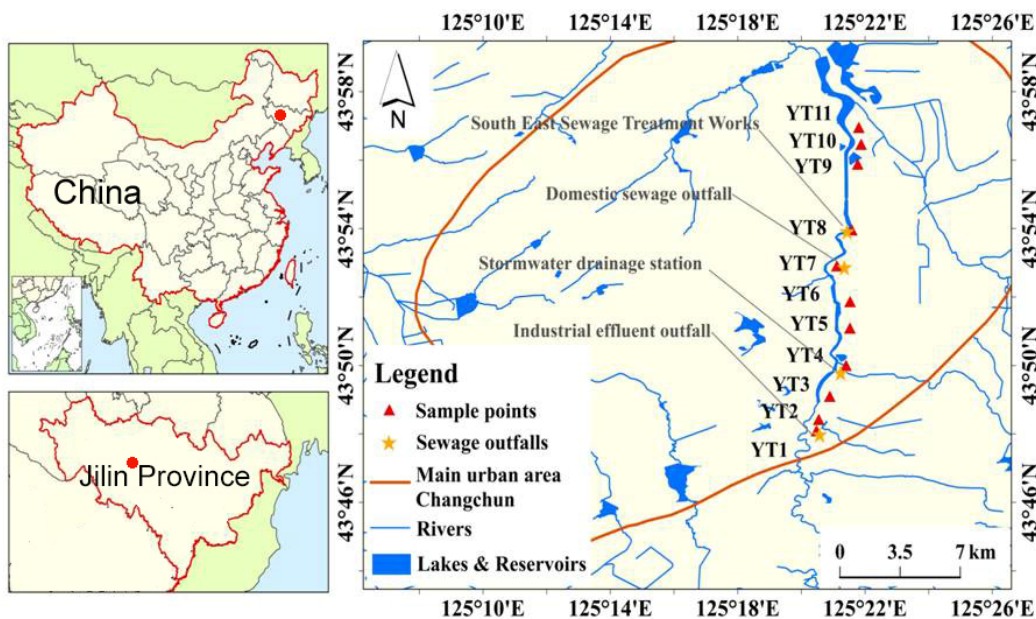

**Figure 1.** Maps of the locations sampled in Yitong River, Changchun, and Urban sewage outfall.

### 2.3. In Situ Sample Collection and Processing

Surface water samples from the Yitong River were collected at 11 stations, and a global positioning system was used to record these stations' coordinates (Figure 1). These riverine sites were regularly monitored before and after three typhoons, and sampling occurred on 25 August, 28 August, 5 September, and 10 September 2020, respectively. Rinse Perspex water sampler, Niskin bottles, and brown glass bottles were employed. We collected water samples in 5 L brown glass bottles at 0–0.2 m below the surface at sites along the river. Following collection, the water samples were transported to the lab for immediate treatment and kept at 4 °C in a cooler. In brief, about 1 L each for each water sample was filtered with glass fiber filters (GF/C, Whatman, Maidstone, United Kingdom, 1.2 μm pore size).

### 2.4. Extraction of PAHs

Solid-phase extraction (SPE) cartridges system (Cleanert S C18-SPE, 6 ML, Agela Technologies Inc., Torrance, CA, USA), equipped with C18 cartridges, was used to extract PAHs from filtered water samples. Before the extraction, the C18 cartridges were prewashed with 10 mL dichloromethane, 10 mL methanol, and 10 mL deionized water (added in order; the above steps are all natural filtration). Then, the filtered water samples were passed through prewashed cartridges at a constant flow rate of 3–5 mL min$^{-1}$ under vacuum pump. After that, these cartridges were added 15 mL of deionized water to remove matrix interference. Then, these cartridges were dried as possible (about 5–10 min). After extraction, the PAHs trapped were eluted to conical flasks by the eluent 1:1:1 (*v/v/v*) dichloromethane, acetone, and normal hexane. The eluent was added to conical flasks using a pipette gun before it was evaporated by gentle stream of nitrogen (nitrogen blowing apparatus N-EVAP-24). Finally, the extract was concentrated to 2 mL and analyzed by GC–MS.

### 2.5. GC–MS Analysis and Quality Control

In this study, the standards and extracts of the PAHs were analyzed using an Agilent 7820 A gas chromatograph equipped with an Agilent 5977 A mass spectrometer (GC–MS) (Agilent Technologies, Avondale, PA, USA), operating in electron impact and selective ion

monitoring modes. The 1 μL extracts were injected into the GC–MS in splitless mode with inlet temperature of 250 °C and with the chromatographic column (Agilent 19091s-433UI HP-5 ms, 30 m × 0.25 mm × 0.25 m), and its flow rate was 1.5 mL/min. Likewise, the temperature programming started from 50 °C to 150 °C with 20 °C/min and stays for 2 min; it continued to rise to 290 °C with 12 °C min$^{-1}$ and was kept for 8 min. The temperature of ion source was 230 °C, and the temperature of quadrupole rod was 150 °C, respectively.

The external standard and calibration curve method based on the six-point calibration curves of the individual PAHs, prepared at concentrations ranging from 5 to 2000 μg/mL (1000, 500, 200, 100, 50 and 5), was used for the quantitative analysis. A typical mixture of the 16 USEPA-listed priority PAHs (200 μg/mL) dissolved in methanol/DCM (1:1, *v/v*) was used and stored at 4 °C. More details of the samples plots, their parallel samples, and standard curve can be found in Supplementary Material.

Instrumental limits of detection (LOD) were calculated from the signal-to-noise ratio of 3 for the pure standard solutions injected into the column. The LOD value was determined based on the standard deviations of three replicate analyses, using the lowest calibration standard. The calculated LOD (mean) values were 1 ng/L (for Naphtalene, Acenaphthyle, Acenaphthene, Fluorene, Phenanthrene, Anthracene, Fluoranthene, Pyrene, Benzo(a)anthracene, Benzo[*c*]phenanthrene) and 2 ng/L (for Benzo(b)fluoranthene, Benzo(k)fluoranthene, Benzo(a)pyrene, Indeno[1,2,3-*cd*]pyrene, Dibenzo(a,h)anthracene, Benzo[*ghi*]perylene), respectively. Measurements below the LOD were set equal to zero. Procedural blanks and spiked samples were processed along with each extraction round of 10 samples. No target PAHs was found in procedural blanks. The spike recoveries were measured in preliminary assays where 3 ng/L of each standard were added to the duplicate samples. The mean recoveries were Naphtalene, 70.7%; Acenaphthyle, 107.1%; Acenaphthene, 88.4%; Fluorene, 91.1%; Phenanthrene, 75.2%; Anthracene, 112.1%; Fluoranthene, 87.8%; Pyrene, 86.6%; Benzo(a)anthracene, 139.8%; Benzo[*c*]phenanthrene, 111.4%; Benzo(b)fluoranthene, 124.4%; Benzo(b)fluoranthene, 103.9%; Benzo(a)pyrene, 127.8%; Indeno[1,2,3-*cd*]pyrene, 113.8%; Dibenzo(a,h)anthracene, 192.5%; and Benzo[*ghi*]perylene, 133.5%.

*2.6. Toxic Equivalent Quantity for Human Risk*

Models of lifetime cancer risk are frequently used to evaluate the toxicity of PAHs to various populations [32]. Waterborne PAHs enter the body mainly through direct ingestion and dermal absorption [33]. A toxicity equivalence factor (TEF) to determine the toxicity of PAHs monomers was introduced and the toxicity equivalence (TEQ) values [34] were calculated as follows:

$$TEQ = \sum C_i \times TEF_i \tag{1}$$

where $C_i$ is the concentration of a specific PAH and $TEF_i$ is the PAH's specific TEQ factor. The toxicity equivalent factor for each compound is shown in Table 1. BaP was set to 1 and the TEF values for the other 15 PAHs monomers were derived by comparing the magnitude of toxicity with an equivalent amount of BaP. The carcinogenic risk index (R) for PAHs was calculated as follows:

$$R = \frac{TEQ_{BaP} \times IR \times CSF \times EF \times ED}{BW \times AT} \tag{2}$$

$$TEQ_{BaP} = \sum_{i=1}^{n} C_i \times TEF_i \tag{3}$$

where EF represents the number of days of exposure per year (d/a) and was taken as 365; CSF is the carcinogenic slope factor and was taken as 7.3 kg/d/mg in this study [35–37]; ED is the exposure time and was taken as 74 a and 78 a for adult males and females, respectively, and 12 a for children; AT is the time and was calculated from (ED × EF); BW is the body weight (kg), which was taken as 67.7, 59.6, and 26.8 in this study; and $C_i$ is the detected concentration of each individual PAHs (ng/L).

**Table 1.** RQs for maximum permissible concentrations (MPCs) and negligible concentrations (NCs) for the individual and total PAHs.

| PAHs | TEF | NCs (ng/L) | MPCs (ng/L) | PAHs | TEF | NCs (ng/L) | MPCs (ng/L) |
|------|-----|-----------|-------------|------|-----|-----------|-------------|
| Nap | 0.001 | 12 | 1200 | BaA | 0.1 | 0.1 | 10 |
| Acy | 0.001 | 0.7 | 70 | Chr | 0.01 | 3.4 | 340 |
| Ace | 0.001 | 0.7 | 70 | BbF | 0.1 | 0.1 | 10 |
| Flu | 0.001 | 0.7 | 70 | BkF | 0.1 | 0.4 | 40 |
| Phe | 0.001 | 3 | 300 | BaP | 1 | 0.5 | 50 |
| Ant | 0.01 | 0.7 | 70 | DahA | 0.1 | 0.5 | 50 |
| Fla | 0.001 | 3 | 300 | InP | 1 | 0.4 | 40 |
| Pyr | 0.001 | 0.7 | 70 | BghiP | 0.001 | 0.3 | 30 |
| | | | | $\sum$16PAHs | – | 27.2 | 2720 |

"–" means no data.

### 2.7. Risk Index for Ecological Risk

Kalf et al. [38] proposed that the ecotoxicity of PAHs was assessed using a risk quotient (RQ). The RQs can be calculated using negligible concentrations (NCs) and maximum permissible concentrations (MPCs) for each PAH (Table 1). In this study, seven PAHs monomers were detected and included in the evaluation. RQ values were calculated as follows:

$$RQ = C_{PAHs}/C_{QV} \tag{4}$$

where $C_{PAHs}$ is the concentration of each PAH monomer in the sample (ng/L), calculated as the average concentration of each monomer, and $C_{QV}$ is the risk value (ng/L) corresponding to each monomer. The lowest risk standard concentration values for PAHs ($C_{QV-NCs}$, ng/L) and maximum permissible concentrations ($C_{QV-MPCs}$, ng/L) are commonly used to calculate the risk entropy values, as follows:

$$RQ_{NCs} = C_{PAHs}/C_{QV-NCs} \tag{5}$$

$$RQ_{MPCs} = C_{PAHs}/C_{QV-MPCs} \tag{6}$$

where $RQ_{NCs} < 1$ denotes a negligible ecological risk for a given PAH, $RQ_{NCs} > 1$ and $RQ_{MPCs} < 1$ indicate that the ecological risk of individual PAHs is moderate, and $RQ_{MPCs} > 1$ indicate that the ecological risk of individual PAHs is high.

### 2.8. Statistic Analysis

Statistical analyses using SPSS (version 16.0; IBM Co., Armonk, NY, USA), such as analysis of variance (ANOVA), was used to examine the differences among PAH levels and their ecological risk before and after typhoons. The pollution sources can be determined by principal component analysis (PCA), factor analysis, and diagnostic ratios of the selected PAHs. In our PCA analysis, the reduced dimensionality or covered largest variance, both calculated by the varimax rotation method, were used to enhance the interpretation of PCA results by rotating the axes. The Origin (version 8.0; Origin Inc., Cleveland, OH, USA) was used to plot figures.

### 3. Results and Discussion

#### 3.1. Occurrence of PAHs in Riverine Samples

In Table 2, the levels of 16 PAHs in river water samples collected from the Yitong River are displayed. The remaining PAHs were detected in the water samples except for the 5–6 ring. The phenanthrene (PHE) boasted the highest percentage in each sample collected during the three typhoons, which was 40.47 ng/L (before Bavet), 31.23 ng/L (before Metsak and after Bavet), 16.47 ng/L (before Poseidon and after Metsak), and 10.16 ng/L (after Poseidon), whereas the decrease in PHE proportions ranged from 2.3% to 20.4% for Σ16PAHs. Similarly, in four samples, Σ16PAHs were found to be 756.37 ng/L, 604.86 ng/L, 379.81 ng/L, and 279.20 ng/L, respectively. According to the quantity of

aromatic rings, such as light (2–3 ringed), heavy (5–6 ringed), and so on, the composition of PAHs was examined to further describe the presence of 16 PAHs [22] (Figure 2). The results reveal that before the Bavet typhoon, the 2–3- and 4-ringed PAHs comprised 79.0% and 21.0% of Σ16PAHs; a 9.1% decrease in the 2–3-ringed PAHs proportion during the second sampling before Typhoon Mesek and after Typhoon Bavet; a 19.2% decrease during the third sampling before Typhoon Poseidon and after Typhoon Mesek; and a 51.0 % decrease during the fourth sampling after Poseidon. The concentrations of PAHs and Σ16PAHs in the rainfall samples were stable with a decaying effect (Figure S1).

**Table 2.** The concentration levels of 16 PAHs in riverine samples collected before and after three typhoons (Bavet, Metsak, and Poseidon).

| | Samples Collected Before Bavet | Samples Collected Before Metsak and After Bavet | Samples Collected Before Poseidon and After Metsak | Samples Collected After-Poseidon |
|---|---|---|---|---|
| NAP | 0.20 | 0.38 | 0.92 | 0.24 |
| ANY | n.d. | n.d. | n.d. | n.d. |
| ANAA | n.d. | 0.31 | n.d. | n.d. |
| FLU | 9.50 | 3.29 | 1.10 | 0.32 |
| PHE | 40.47 | 31.23 | 16.47 | 10.16 |
| ANT | n.d. | 1.18 | 0.45 | 1.6 |
| FLT | n.d. | n.d. | n.d. | n.d. |
| PYR | 5.57 | 6.03 | 4.95 | 3.65 |
| BaA | 3.97 | 4.37 | 4.57 | 4.20 |
| CHR | 3.33 | 3.32 | 3.20 | 3.09 |
| BbF | n.d. | n.d. | n.d. | n.d. |
| BkF | n.d. | n.d. | n.d. | n.d. |
| BaP | n.d. | n.d. | n.d. | n.d. |
| IPY | n.d. | n.d. | n.d. | n.d. |
| DBA | n.d. | n.d. | n.d. | n.d. |
| BPE | n.d. | n.d. | n.d. | n.d. |

n.d. represents not detected.

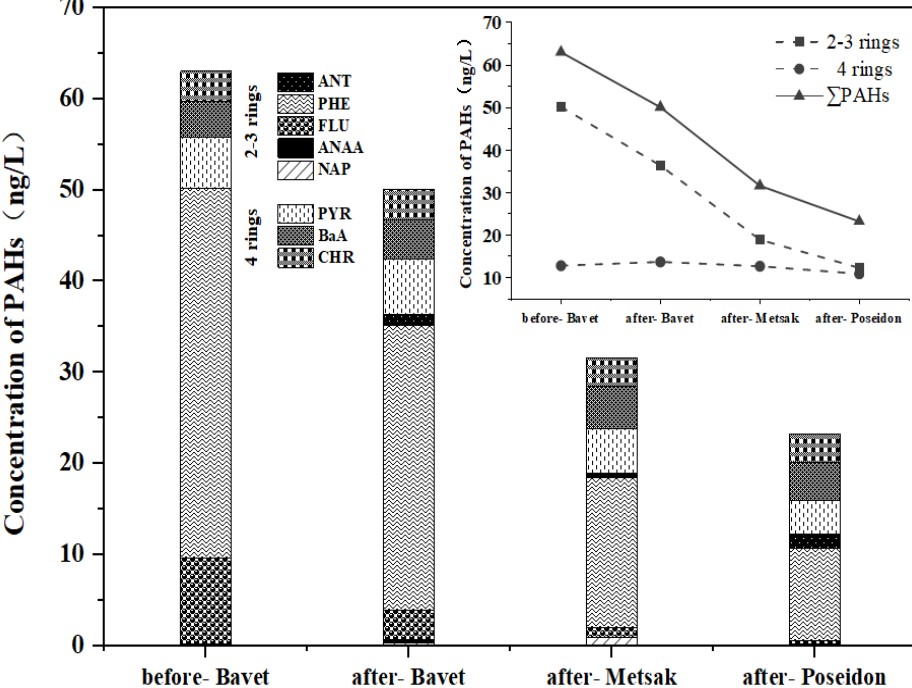

**Figure 2.** The compositional patterns of PAHs during four samplings based on the number of aromatic rings, e.g., light (2–3 ring), 4-ring, and heavy (5–6 ring) PAHs.

### 3.2. Characteristics of 16 PAH Concentrations in Samples of Rainfall

The minimum, maximum, and total concentrations of 16 polycyclic aromatic hydrocarbons in rainwater samples are shown in Table S1. The total concentration of polycyclic aromatic hydrocarbons (PAHs) in precipitation samples after the passage of Typhoon Bavet was 12.20 ng/L, the average concentration was 0.76 ng/L, and the range was 0.64–3.64 ng/L. The 4-ring components made up 71.89% of the total concentration, while the 2–3-ring components made up 28.11% of the total concentration. The total concentration of PAHs collected from Typhoon Metsak precipitation was 28.30 ng/L, with an average concentration of 1.77 ng/L and a range of 0.60 to 15.27 ng/L. Among them, components with 2–3 rings made up 64.1% of the overall concentration, while those with 4 rings made up 35.9%. The 2–3 ring component accounted for 64.1% of the total concentration, while the 4 ring component accounted for 35.9%. The total PAH concentration in Typhoon Poseidon precipitation was 25.53 ng/L, with a mean concentration of 1.59 ng/L and a range of 0.34–13.97 ng/L, with the 2–3 ring component accounting for 56.05% of the total concentration. Phenanthrene concentrations were higher in the second and third sample collections. Moreover, 5–6 ring components were not detected in the three rainfall events, which may be because compared with the same group of polycyclic aromatic hydrocarbons, low-ring polycyclic aromatic hydrocarbons have higher water solubility and volatility and more easily enter the clouds and settle with rain. The results show that this study is similar to the conclusion of Junesoo Park's [39] previous study on atmospheric deposition in Galveston Bay, TX, USA; that is, 2–4 rings are the main polycyclic aromatic hydrocarbons dissolved in rainwater samples, and phenanthrene is one of the most important polycyclic aromatic hydrocarbons in rainwater. In addition, Wei et al. detected a high concentration of 36.9 mg/L PAHs in snow samples collected in Northeast China, which emphasizes the low PAHs level found in this study. This may be due to differences in local pollution levels between the regions studied, as the PAHs emitted accumulate only near the source [40].

### 3.3. Spatial Distributions of PAHs in Riverine Samples

The spatial variations in Σ16PAHs in the Yitong River before and after typhoons are shown in Figure 3, from which it can be seen that the total PAH concentrations at the various sampling sites varied from 15 ng/L to 64 ng/L. Significant differences (F = 48.212, $p < 0.001$) in Σ16PAHs were found among the different sampling sites. Before the Metsak typhoon, riverine Σ16PAH concentrations were relatively high at YT4 and YT6, whereas the concentrations at YT3 were relatively lower (Figure 1). YT3 is the upstream sampling point at Changchun's river outfall, and YT6 is located downstream of the domestic wastewater outfall. Jingkai I and II were all discharged into the Yitong River, resulting in high concentrations of PAHs. In the second sampling (before Metsak and after Bavet), Σ16PAHs were higher in YT8 and relatively lower in YT3 and YT10, in comparison to the PAH levels in the first sampling. YT8 is the East Bridge combined storm drainage station, located west of the Donglai Bridge and on the north bank of the Yitong River. A large amount of domestic sewage and rainwater mixes can overflow into the river through open-channel overflow weirs and sewer wells.

In the third sampling (before Poseidon and after Metsak), Σ16PAHs were high at YT7 and YT9, which are located downstream of the industrial effluent outfall near the CHP II plant and therefore receive more effluent, and Σ16PAHs were low at YT3 and YT10. The results of this spatial distribution of PAHs were consistent with those of the second sampling. In the fourth sampling (post-Poseidon), Σ16PAHs were the highest in YT1 and YT2, and the lowest in YT10. The discharged water from the Southeast Wastewater Treatment Plant near YT1 was returned to the Yitong River as landscape water, which may be the direct cause of the high PAH concentrations in this study. YT2 is located near the South Third Ring Road stormwater outfall, and the initial stormwater pollution is usually complex in composition and high in pollutant concentration.

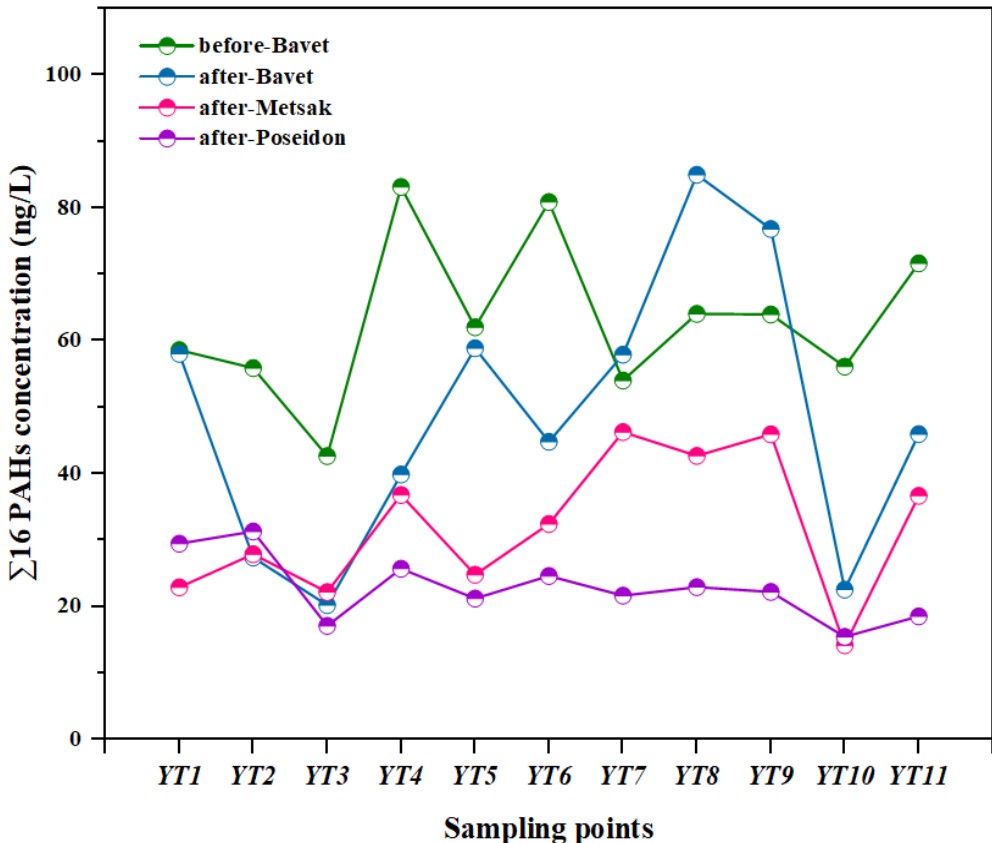

**Figure 3.** Distributions f total 16 PAHs in water samples.

Since 2016, the Changchun government has constructed Nanxi Wetland Park upstream of the urban Yitong River, which is an artificial wetland system. Previous research has shown that wetlands can lessen the pollution caused by urban wastewater [41–43]. In early 2016, construction crews entered the river and removed the old sewer lines while dredging the river. According to this information, the starting point of the old sewage pipeline of the Yitong River is located 100 m upstream of YT6 and ends between YT9 and YT10, with a total length of 13 km for the main line and 11 km for the branch line. Because these pipelines have been in use for nearly 20 years, corrosion is very serious, resulting in the direct leakage of sewage into the Yitong River.

To solve this problem, Changchun implemented interception projects on both banks of the Yitong River between YT6 and YT10, intercepting all the sewage to the sewage pipeline on the banks and removing all the old pipelines in the river, which made a significant contribution to the treatment of the Yitong River's water quality. Therefore, the contamination in samples collected downstream of YT10 was low in these four samples. According to the distribution of PAHs sampled on these four occasions, the pollution of PAHs in the middle section of the Yitong River was more serious, and individual water outlet gates did not meet the requirements of pollution control and were not closed in time after the rain stops; thus, sewage was directly discharged into the river. In addition, because 80% of the built-up area within the middle section of the watershed part of the community was dilapidated and there was rubbish everywhere, the initial rainwater pollution was serious, and the polluted area was discharged directly into the Yitong River through pipes.

### 3.4. Identification of PAH Sources

3.4.1. Sources by Diagnostic Ratios of Selected PAHs

There are numerous ways to locate the sources of PAHs [44], and certain PAH diagnostic ratios, such as ANT/(ANT + PHE), are straightforward and widely used [45]. When the ANT/(ANT + PHE) ratios are below 0.1, it is possible to identify petroleum

sources, and combustion sources when the ratios are above 0.1. For our riverine samples (Figure 4), the ANT/(ANT + PHE) ratio was determined to be 0 before Bavet (first sampling), 0.036 before Metsak and after Bavet (second sampling), and 0.027 before Poseidon and after Metsak (third sampling), indicating the dominant role of petroleum sources. Similarly, the ANT/(ANT + PHE) ratio was 0.136 after Poseidon, signifying that PAHs in these samples were derived from thermal sources.

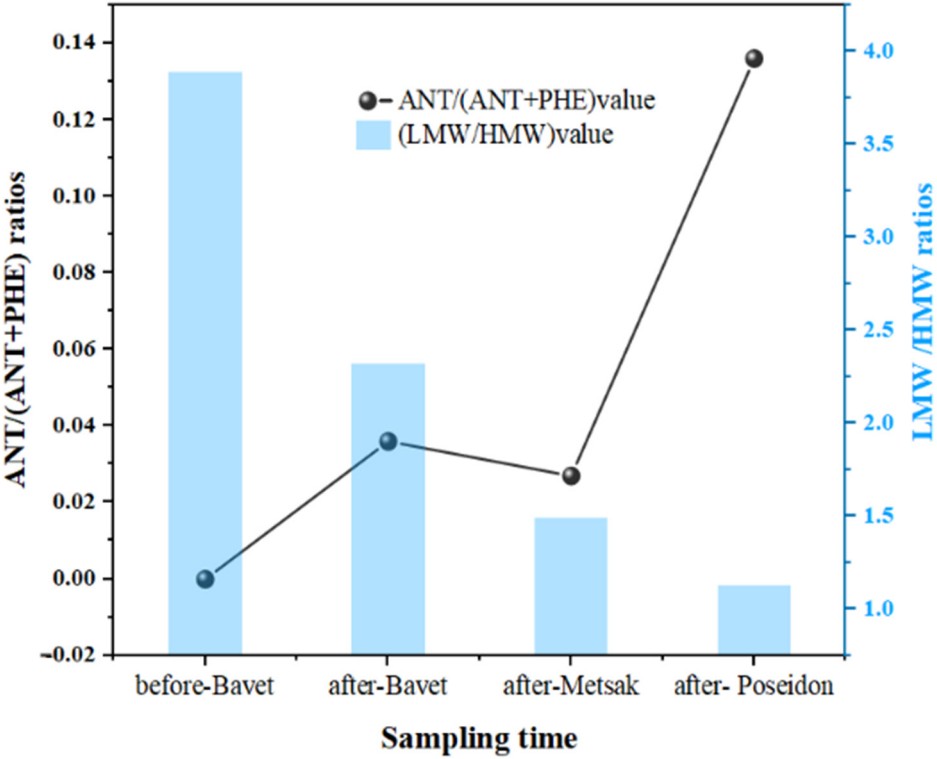

**Figure 4.** ANT/(ANT + PHE) ratios and LMW/HMW ratios for 16 PAHs.

Additionally, it is possible to determine the sources of PAHs using their isomers, introduced by LMW (LMW, 2 and 3 rings) and HMW (HMW, >4 rings); that is, LMW/HMW < 1 implies a combustion source of PAHs, whereas LMW/HMW > 1 implies a petroleum source [43]. As shown in Figure 4, our results indicate that the LMW/HMW of the riverine samples was >1, signifying that PAHs mainly derived from the low-temperature transformation of organic matter and oil spills.

### 3.4.2. Sources by PCA

The first three factors accounted for >70% of the variability. Considering the outcomes of a prior investigation [46–49], diagnostic PAH compounds from different sources were selected. Flu was used as a diagnostic PAH compound to quantify coke oven emissions; Phe, Pyr, Ant, and Chr were used to quantify PAH compounds originating from coal combustion; and BaA and Acy were considered to reflect natural gas and fuelwood biomass combustion, respectively, whereas Nap denoted the source as oil spills. BbF in the environment is mainly derived from the combustion of coal and oil. The results of the principal component analysis (Figure 5) showed that principal components 1, 2, and 3 accounted for 38.4%, 19.2%, and 12.9% of the variability in the Yitong River water samples, respectively. Factor 1 contains high concentrations of Chr, Pyr, and BbF, which are typically present in high concentrations of coal combustion discharges; therefore, the first principal component can be considered to represent a coal combustion source. Factor 2 contains high concentrations of BbF and BaA, where BaA dominates the natural gas combustion signal; therefore, factor 2 is considered to represent the contribution of coke oven and gasoline combustion. Factor 3 is dominated by Nap and Ant, reflecting contributions from coal combustion and oil spills.

Our results therefore suggest that PAHs in river waters are contaminated by a mixture of water bodies contaminated from coal and motor vehicle petroleum combustion emissions and petroleum spills.

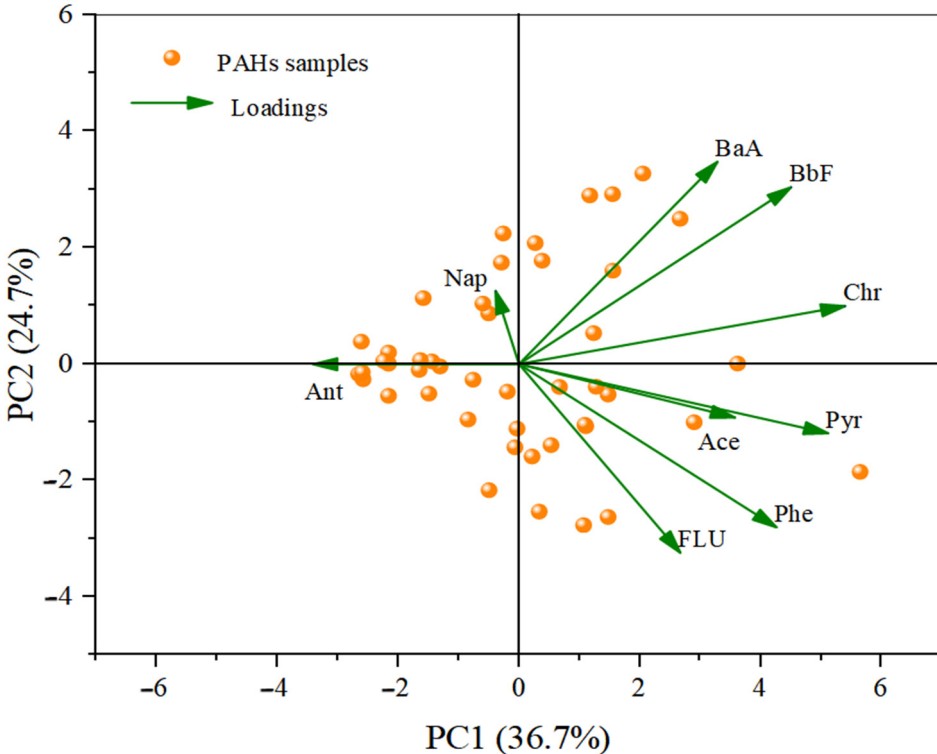

**Figure 5.** Principal component analysis of data from the Yitong River and PAHs samples. Score plot First and second principal components.

*3.5. Environmental Risk Assessment*

3.5.1. Human Health Risk Assessment

In the first sampling (before the Metsak typhoon) (Table 3), the carcinogenic risk of PAHs espousing for adult males, adult females, and children in riverine samples during typhoons were in the ranges of $3.99 \times 10^{-5}$–$7.88 \times 10^{-2}$, $4.53 \times 10^{-5}$–$8.97 \times 10^{-2}$, and $5.45 \times 10^{-5}$–$2.16 \times 10^{-2}$, respectively. This suggests a high carcinogenic risk of PAH in children. The order of magnitude of BaA was also found to be higher, in the range of $10^{-1}$–$10^{-2}$, indicating a higher risk than those of other PAHs, followed by Phe, Pyr, and Chr with $10^{-3}$ order of magnitudes.

**Table 3.** TEQ for each PAHs.

| Sampling Time | PAHs | TEQ | | |
|---|---|---|---|---|
| | | Men | Women | Children |
| Before Bavet (first sampling) | NAP | $3.99 \times 10^{-5}$ | $4.53 \times 10^{-5}$ | $1.09 \times 10^{-5}$ |
| | ANAA | – | – | – |
| | FLU | $1.89 \times 10^{-3}$ | $2.15 \times 10^{-3}$ | $5.17 \times 10^{-4}$ |
| | PHE | $8.07 \times 10^{-3}$ | $9.17 \times 10^{-3}$ | $2.20 \times 10^{-3}$ |
| | ANT | – | – | – |
| | PYR | $1.10 \times 10^{-3}$ | $1.26 \times 10^{-3}$ | $3.03 \times 10^{-4}$ |
| | BaA | $7.88 \times 10^{-2}$ | $8.97 \times 10^{-2}$ | $2.16 \times 10^{-2}$ |
| | CHR | $6.62 \times 10^{-3}$ | $7.55 \times 10^{-3}$ | $1.81 \times 10^{-3}$ |

**Table 3.** *Cont.*

| Sampling Time | PAHs | TEQ | | |
|---|---|---|---|---|
| | | **Men** | **Women** | **Children** |
| After Bavet (second sampling) | NAP | $7.59 \times 10^{-5}$ | $8.61 \times 10^{-5}$ | $2.07 \times 10^{-5}$ |
| | ANAA | $6.18 \times 10^{-5}$ | $7.02 \times 10^{-5}$ | $1.69 \times 10^{-5}$ |
| | FLU | $6.56 \times 10^{-4}$ | $7.45 \times 10^{-4}$ | $1.79 \times 10^{-4}$ |
| | PHE | $6.23 \times 10^{-3}$ | $7.08 \times 10^{-3}$ | $1.70 \times 10^{-3}$ |
| | ANT | $2.35 \times 10^{-3}$ | $2.67 \times 10^{-3}$ | $6.43 \times 10^{-4}$ |
| | PYR | $1.19 \times 10^{-3}$ | $1.37 \times 10^{-3}$ | $3.29 \times 10^{-4}$ |
| | BaA | $8.69 \times 10^{-2}$ | $9.90 \times 10^{-2}$ | $2.38 \times 10^{-2}$ |
| | CHR | $6.60 \times 10^{-3}$ | $7.52 \times 10^{-3}$ | $1.80 \times 10^{-3}$ |
| After Metsak (third sampling) | NAP | $1.84 \times 10^{-4}$ | $2.08 \times 10^{-4}$ | $5.01 \times 10^{-5}$ |
| | ANAA | – | – | – |
| | FLU | $2.19 \times 10^{-4}$ | $2.49 \times 10^{-4}$ | $5.99 \times 10^{-4}$ |
| | PHE | $3.29 \times 10^{-3}$ | $3.73 \times 10^{-3}$ | $8.97 \times 10^{-4}$ |
| | ANT | $8.96 \times 10^{-4}$ | $1.01 \times 10^{-3}$ | $2.45 \times 10^{-4}$ |
| | PYR | $9.85 \times 10^{-4}$ | $1.04 \times 10^{-3}$ | $2.69 \times 10^{-4}$ |
| | BaA | $9.09 \times 10^{-2}$ | 0.104 | $2.49 \times 10^{-2}$ |
| | CHR | $6.36 \times 10^{-3}$ | $7.25 \times 10^{-3}$ | $1.74 \times 10^{-3}$ |
| After Poseidon (forth sampling) | NAP | $4.79 \times 10^{-5}$ | $5.44 \times 10^{-5}$ | $1.31 \times 10^{-5}$ |
| | ANAA | – | – | – |
| | FLU | $6.38 \times 10^{-5}$ | $7.25 \times 10^{-5}$ | $1.74 \times 10^{-5}$ |
| | PHE | $2.02 \times 10^{-3}$ | $2.30 \times 10^{-3}$ | $5.53 \times 10^{-4}$ |
| | ANT | $3.18 \times 10^{-3}$ | $3.63 \times 10^{-3}$ | $8.72 \times 10^{-4}$ |
| | PYR | $7.26 \times 10^{-4}$ | $8.27 \times 10^{-4}$ | $1.99 \times 10^{-4}$ |
| | BaA | $8.35 \times 10^{-2}$ | $9.51 \times 10^{-2}$ | $2.29 \times 10^{-2}$ |
| | CHR | $6.15 \times 10^{-3}$ | $7.00 \times 10^{-3}$ | $1.68 \times 10^{-3}$ |

"–" indicates non-detected PAH concentrations.

In the second sampling (before Metsak and after Bavet), the carcinogenic risk of PAHs exposure for adult males, adult females, and children ranged from $6.18 \times 10^{-5}$ to $8.69 \times 10^{-2}$, $7.02 \times 10^{-5}$ to $9.9 \times 10^{-2}$, and $1.69 \times 10^{-5}$ to $2.38 \times 10^{-2}$, respectively. The PAHs detected were all carcinogenic, with the highest risk being BaA at the $10^{-2}$ order of magnitude level and the lowest being NaP at the $10^{-5}$ order of magnitude level. A class of PAHs not present before the typhoon—namely, ANA—was detected after Typhoon Bavet's landfall and may have been carried by rainfall, with concentrations at $10^{-5}$ levels, also posing some carcinogenic risk to humans. Notably, FLU risks at $10^{-4}$ levels, PHE ($10^{-3}$ to $10^{-4}$), PYR ($10^{-3}$), and CHR ($10^{-3}$) pose a non-negligible carcinogenic risk. The overall carcinogenic risk of PAHs in the water column was marginally lower than before the typhoon.

In the third sampling (before Poseidon and after Metsak), carcinogenic risk was $1.84 \times 10^{-4}$–$9.09 \times 10^{-2}$, $2.08 \times 10^{-4}$–$1.04 \times 10^{-1}$, and $1.31 \times 10^{-5}$–$2.49 \times 10^{-2}$ for adult males, adult females, and children, respectively. The highest risk in the sampling environment was still BaA, and the carcinogenic risk was higher than that before Typhoon 8 and the first sampling, probably due to some of the BaA carried by rainwater into the river. The relatively low carcinogenic risk ranges of PYR, PHE, and FLU were all in the

order of $10^{-3}$ to $10^{-4}$, and the lowest carcinogenic PAH species was NaP, which was in the $10^{-4}$–$10^{-5}$ range.

The range of PAHs for adult males, adult females, and children affected by the fourth sampling (post-Poseidon) was $4.79 \times 10^{-5}$–$8.35 \times 10^{-2}$, $5.44 \times 10^{-5}$–$9.51 \times 10^{-2}$, and $1.31 \times 10^{-5}$–$2.29 \times 10^{-2}$, respectively. Compared to the typical cancer-causing effects of PAHs in the water column of the two prior typhoons, the effect of rainfall scouring was more significant, probably due to the fact that Typhoon Neptune 10 had a longer duration of rain and wind than Typhoon Metsak 9, starting in the evening of 7th September and continuing until 9th September. The PAH with the highest carcinogenic risk was of BaA at $10^{-2}$, followed by PHE, ANT, and CHR at $10^{-3}$, and slightly lower environmental carcinogenic risks of FLU and NAP were found, both at $10^{-5}$ orders of magnitude.

Consistent with a previous study [50], the risk was higher in children than in adults, probably due to the higher exposure of children to pollutants. In summary, the carcinogenic risk of single PAHs was measured in all four sampling sessions before and after typhoon transit in this study, and the carcinogenic risk range of BaA was 2–3 orders of magnitude higher than that of the other species. Further comprehensive investigation of the distribution and toxicological properties of the contaminants is needed to better protect drinking water health and water ecological safety.

### 3.5.2. Ecological Risk Assessment

As shown in Table 4, the RQMPCs of all PAHs detected in the river samples were <1 during the four sampling sessions before and after the typhoon, whereas the RQNCSs of all PAHs, except NaP and CHR, were >1. The ecological risk of the five PAHs in the river samples was negligible, and ANT and CHR were of moderate ecological risk. Notably, the RQMPCs of BaA had a higher ecological risk, with 1–2 orders of magnitude compared to the other PAHs. Adverse ecological risks for an individual PAH rarely occurred, and the overall level of ecological risk for PAHs in the waters of the Yitong River before and after typhoon transit was low.

**Table 4.** Risk quotients for 16 PAHs ($\mu$g/L) in the water.

| PAHs. | Samples Collected Before Bavet | | Samples Collected Before Metsak and After Bavet | | Samples Collected Before Poseidon and After Metsak | | Samples Collected After Poseidon | |
|---|---|---|---|---|---|---|---|---|
| | RQ$_{NCs}$ | RQ$_{MPCs}$ | RQ$_{NCs}$ | RQ$_{MPCs}$ | RQ$_{NCs}$ | RQ$_{MPCs}$ | RQ$_{NCs}$ | RQ$_{MPCs}$ |
| Nap | 0.02 | $1.67 \times 10^{-4}$ | 0.03 | $3.17 \times 10^{-4}$ | 0.08 | $7.67 \times 10^{-4}$ | 0.02 | $2 \times 10^{-4}$ |
| Acy | – | – | – | – | – | – | – | – |
| Ace | – | – | 0.44 | $4.40 \times 10^{-3}$ | – | – | – | – |
| Flu | 13.57 | 0.14 | 4.70 | 0.05 | 1.57 | 0.02 | 0.46 | $4.57 \times 10^{-3}$ |
| Phe | 13.49 | 0.13 | 10.41 | 0.10 | 5.49 | 0.05 | 3.39 | 0.03 |
| Ant | – | – | 1.69 | 0.02 | 0.64 | 0.01 | 2.29 | 0.02 |
| Fla | – | – | – | – | – | – | – | – |
| Pyr | 7.96 | 0.08 | 8.61 | 0.09 | 7.07 | 0.07 | 5.21 | 0.05 |
| BaA | 39.70 | 0.40 | 43.70 | 0.44 | 45.70 | 0.46 | 42.00 | 0.42 |
| Chr | 0.98 | 0.01 | 0.98 | $9.76 \times 10^{-3}$ | 0.94 | $9.41 \times 10^{-3}$ | 0.91 | $9.09 \times 10^{-3}$ |
| BbF | – | – | – | – | – | – | – | – |
| BkF | – | – | – | – | – | – | – | – |
| BaP | – | – | – | – | – | – | – | – |
| DahA | – | – | – | – | – | – | – | – |
| InP | – | – | – | – | – | – | – | – |
| BghiP | – | – | – | – | – | – | – | – |
| $\sum_{16}$PAHs | 75.72 | 0.76 | 68.87 | 0.69 | 60.85 | 0.61 | 51.99 | 0.51 |

"–" indicates non-detected PAH concentrations.

### 3.6. PAHs Characteristics and Risk before and after Typhoon

The three heavy precipitation events clearly resulted in a gradual decrease in total PAH concentrations in the Yitong River. The $\Sigma$16PAH concentrations varied significantly before and after the onset of typhoons, showing a decreasing trend with rainfall flushing and a gradual decrease in the proportion of low-ring PAHs. This can be attributed to the dilution effect of drainage and rainfall [51,52]. Numerous studies on the dynamics of PAH levels in aquatic environments have been carried out—for example, the Cross River in southeastern Nigeria [53], the Gaoping River in Taiwan [51], and the Pearl River in China [54]—and their results showed that dynamic PAH levels are strongly related to changes in river discharge and rainfall. The differences between the three typhoons were also compared, and the ANOVA results showed that there were no significant differences (F = 0.96, $p > 0.05$) in $\Sigma$16PAH levels.

The following differences between the ecological risk of PAHs in water before and after the typhoon's passage were found: after Typhoon 8, the ecological risk values of FLU, PHE, PYR, and BaA to the environment increased, whereas NaP increased slightly; after Typhoon No. 9, the ecological risk of all the PAHs detected a decreased trend except for NaP, in addition to the detection of ANAA whose ecological risk value was lower in this sampling; whereas in the last sampling after Typhoon No. 10, Poseidon, the human health risk of all six PAHs detected decreased except for ANT. The changes in $\Sigma$16PAHs before and after the typhoons showed a decreasing trend, as did their ecological risk values ($RQ_{NCs}$ and $RQ_{MPCs}$).

The human health risks of PAHs in riverine water before and after the three typhoons were as follows: after Typhoon Bawai, the health risks of NaP, ANAA, ANT, PYR, and BaA for men, women, and children increased, whereas the health risks of FLU, PHE, and CHR for humans decreased; after Typhoon Metsak, all five PAHs were detected, except for NaP and BaA, which showed decreased health risks to humans; in addition, in the last sampling after Typhoon Poseidon, all six PAHs were detected, except ANT, and the health risks showed a decreasing trend in humans. The health risk values for all six PAHs were found to have decreased. The human health risks of PAHs were consistent with the PAH concentrations, with high concentrations at the beginning of the typhoons, which stabilized as the rainfall progressed.

## 4. Conclusions

Overall, in this study, fundamental information about the distribution characteristics and potential sources of PAHs in urban riverine waters located in Changchun before and after the three super typhoons in 2020 was presented. Only 7 of the 16 PAHs were detected in the study area, and no 6-ringed PAHs were detected. LMW-PAHs were present in significant quantities in the water samples. The total PAH concentrations showed a significant decrease after the typhoons, owing to the scouring and dilution effects of rainwater. Similarly, the ratio characterization method and PCA revealed that a variety of sources, including coal and motor vehicle petroleum combustion emissions, were responsible for the PAH pollution in the waters, of which some contribution was from petroleum spills. In addition, exposure to PAHs caused by typhoons poses a significant health risk to humans, but the risk to children is slightly higher than that to adults. From the perspective of ecological sustainability and environmental safety, necessary measures should be considered to prevent further PAH pollution.

**Supplementary Materials:** The following supporting information can be downloaded at: https://www.mdpi.com/article/10.3390/su15075777/s1, Figure S1: Distribution of polycyclic aromatic hydrocarbon concentrations in precipitation samples from three super typhoons, "Bavet", "Metsak" and "Poseidon"; Table S1: PAHs concentrations in rainwater samples from Typhoon "Bavet", "Metsak" and "Poseidon". Refs. [55–57] are cited in Supplementary Materials.

**Author Contributions:** G.M. and M.Z. were responsible for analysis of data and writing—original draft. D.B. and X.W. were responsible for writing—review and editing, funding acquisition. G.M. and F.C. were responsible for collection of riverine water samples and in-house experimental analysis. All authors have read and agreed to the published version of the manuscript.

**Funding:** This study was supported by the Young Development Technology Project awarded to Mu from Science and Technology Department of Jilin Province (Grant No. 20230508123RC).

**Data Availability Statement:** The data presented in this study are available on request from corresponding author.

**Acknowledgments:** The authors would like to thank all staff and students for their unfailing help with field sampling and laboratory analysis.

**Conflicts of Interest:** The authors declare no conflict of interest.

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
