# Peer review of "Pollution and Risk Assessment of Polycyclic Aromatic Hydrocarbons in Urban Rivers in a Northeastern Chinese City: Implications for Continuous Rainfall Events"

_sustainability, doi:10.3390/su15075777_

Round 1

Reviewer 1 Report

The authors tested PAHs concentrations during three typhoon events, and analyzed the concentration changes during this period. More detailed information and discussion are needed to show the novelty, significance and implications of the research.

1. The research objectives and novelty of the paper are suggested to be emphasized in Introduction. And the introduction can be improved overall.

2. The characteristics of the typhoon events should be provided, not only name, time, and precipitation amount. It is important for readers to know the impact of typhoons.

3. What are the normal background concentrations before the typhoon events?

4. Line 40, minor mistake. Please check the manuscript carefully overall.

5. Sections 2.3 and 2.5 may be combined.

6. Figure 2, the time or typhoon names is suggested for x-axis, not first, second. It may be clearer to readers to understand the relationship.

7. Figure 3, the four figures have same Y-axis, can they be shown in one figure to show the changes?

8. The authors analyzed the source of PAHs for many sites, but what is the specific relationship between PAHs concentrations and typhoons? What are the differences with general rainfall?

Reviewer 2 Report

In this manuscript "Pollution and Risk Assessment of Polycyclic Aromatic  Hydrocarbons in urban rivers: Implications for Extreme Typhoons Events" the authors evaluated  surface water  from urban rivers before and after  typhoons; concentartions of 16 priority PAHs  in riverine samples; and their levels, distribution, toxic equivalent (TEQ). This paper is interesting and suitable for the topic of the journal but needs to be improved in some aspects. My opinion is to accept with minor revisions:

Minor points:

1) Introduction Section: Clarify the objectives of the study at the end of this section

2) Materials and Methods: Enter LOQ and LOD, also report method validation information (precision, repeatability, calibration curve points).

3) Report a table with the target ions of the analyzed compounds

4) 2.8 Statistic analysis: Write more information about the technique used in this study

5) Report a Figure with the results obtained from the PCA

6) One reference at the end of a sentence is not enough, add more references

Reviewer 3 Report

The authors are studying the pollution and risk assessment of polycyclic aromatic hydrocarbons in urban rivers before and after extreme typhoon events. The experimental design and results presented looks reasonable, however, there are some points that needed to be clear before further consideration-

1. the title is focused on the urban rivers, but the location of the study is only focused on Changchun city. This title is too large, the author should consider the specific title base on their study locations;

2. it could be nice if the author could provide some schemes and figures for the water sample pretreatment purification and concentration process, it could be nice;

3. for the GC-MS analysis, the authors provide some of the chromatographies and spectra in the supplementary material to support their results;

4. in section 2.5, the authors mentioned sample volume was 1L, please clarify what sample volume means here, and please also clarify what's your sample concentration factor and the injection volume. More details are needed for your analytical measurement section.

Reviewer 4 Report

In this study, Gas chromatography-mass spectrometry (GC-MS) was used to measure the concentrations of 16 major PAHs in stormwater and water samples from the Yitong River,and it also analyzes the sources and hazards of PAHs. This study has great research value, but some modifications are needed to improve the quality of the manuscript to meet the requirements of the journal.

1. English language needs to be further improved to enhance the quality of the manuscript.

2. It is recommended to add the analysis of Figure S1 to the manuscript, which is helpful for readers to understand the content of the manuscript.

3. Suggest adding the source of Figure 2 to the figure caption.

4. It is recommended to modify some Tables to Figures, which are conducive to intuitively expressing the changes of data, such as changing Table 3 into Figures.

Round 2

Reviewer 1 Report

The authors have responded the reviewer's comments and improved the manuscript.

Author Response

Thank you very much for your valuable guidance and advice.

Reviewer 3 Report

The authors did a good job of revising the manuscript, however, some points still recommended to be adjusted before accepted

1. in the method section, please add the material and chemicals section;

2. for the GC-MS figures provided in the supplementary material, please add more detail, point out which one is the peak for your analytes

3. extensive English language editing is needed, such as on page 3 line 132- water quality should not be described as weak, "poor" might be more appropriate.
